# Dual Relief: How Atopic Dermatitis Treatments Affect Alopecia Areata—A Small Retrospective Cohort Study

**DOI:** 10.3390/diagnostics15050520

**Published:** 2025-02-20

**Authors:** Daciana Elena Brănișteanu, Antonia-Elena Huțanu, Daniel Constantin Brănișteanu, Cristina Colac-Boțoc, Roxana Paraschiva Ciobanu, Cătălina-Anca Munteanu, Alin Gabriel Colac, George Brănișteanu, Cătălina Onu-Brănișteanu, Nicuța Manolache, Mihaela-Paula Toader, Elena Porumb-Andrese

**Affiliations:** 1Discipline of Dermatology, Grigore T. Popa University of Medicine and Pharmacy, 16 Universitatii Str., 700115 Iasi, Romaniaelena.andrese1@umfiasi.ro (E.P.-A.); 2Railways Universitary Hospital, Dermatology Clinic, 1 Garabet Ibraileanu Str., 700115 Iasi, Romania; 3Discipline of Ophtalmology, Grigore T. Popa University of Medicine and Pharmacy, 16 Universitatii Str., 700115 Iasi, Romania; daniel.branisteanu@umfiasi.ro; 4Railways Universitary Hospital, Ophtalmology Clinic, 1 Garabet Ibraileanu Str., 700115 Iasi, Romania; 5Department of Surgery, Oral and Maxillofacial Surgery, Faculty of Dental Medicine, Grigore T. Popa University of Medicine and Pharmacy, 700115 Iasi, Romania; 6Recovery Hospital, Orthopedy Clinic, 14 Pantelimon Halipa Str., 700661 Iasi, Romania; 7Institute for Cardiovascular Diseases C.C. Iliescu, 258 Fundeni Str., 022328 Bucharest, Romaniamihaela.toader@umfiasi.ro (M.-P.T.); 8Department of Pharmaceutical Sciences, Faculty of Medicine and Pharmacy, “Dunarea de Jos” University, 800008 Galati, Romania; 9Discipline of Oral Medicine, Oral Dermatology, Grigore T. Popa University of Medicine and Pharmacy, 16 Universitatii Str., 700115 Iasi, Romania

**Keywords:** atopic dermatitis, alopecia areata, JAK inhibitors, dupilumab, atopic eczema

## Abstract

**Background/Objectives:** Atopic dermatitis (AD) and alopecia areata (AA) frequently coexist due to shared immune-mediated mechanisms. Treatments targeting AD, including Janus kinase (JAK) inhibitors and dupilumab, may impact AA outcomes in unpredictable ways. This study aims to evaluate the effects of advanced therapies on patients with concurrent AD and AA to inform treatment strategies. **Methods:** A retrospective cohort study was conducted on six patients diagnosed with both AD and AA. Treatments included systemic corticosteroids, dupilumab, and JAK inhibitors (baricitinib and upadacitinib). Outcomes were assessed at six months using the Severity of Alopecia Tool (SALT), Dermatology Life Quality Index (DLQI), and Scoring Atopic Dermatitis (SCORAD) scores. **Results:** Patients receiving JAK inhibitors showed significant improvements in AD and AA outcomes, with mean reductions of 95.65% in SALT scores, 91.03% in DLQI scores, and 89.57% in SCORAD scores. Dupilumab was associated with the onset or worsening of AA in two patients. Systemic corticosteroids provided short-term benefits but are unsuitable for long-term management due to safety concerns. **Conclusions:** JAK inhibitors are effective for managing concurrent AD and AA, offering substantial improvements in disease control and quality of life. However, dupilumab requires cautious use in patients with these comorbid conditions. Personalized treatment strategies, informed by patient-specific factors, are essential for optimizing outcomes and minimizing risks. Further research is needed to identify predictive markers and refine therapeutic approaches for this challenging population.

## 1. Introduction

Atopic dermatitis (AD), also known as atopic eczema, is a chronic, relapsing inflammatory skin condition characterized by intensely pruritic, erythematous, and scaly lesions. In adults, these lesions typically appear on flexural surfaces, while in infants, they more frequently affect the face, scalp, trunk, and extensor surfaces. The disease progresses through distinct phases: in the acute phase, lesions present as vesicular eruptions with open, weeping sores and crusting; in the subacute phase, they transform into dry, scaly fissures; and in chronic stages, scratching and repeated inflammation often lead to thickened, lichenified skin with reduced erythema compared to acute flares [1].

AD is prevalent in about 10–20% of children and 1–3% of adults worldwide, with variation across age groups and geography. It frequently coexists with other atopic conditions such as asthma and allergic rhinitis, highlighting an underlying immunologic connection. However, AD is increasingly recognized as a multisystem disorder associated with metabolic and cardiovascular comorbidities, as well as psychological conditions like anxiety and depression [2].

Atopic dermatitis (AD) is traditionally categorized into extrinsic and intrinsic subtypes. The extrinsic phenotype, accounting for approximately 80% of cases, is characterized by elevated serum immunoglobulin E (IgE) levels, Th2-skewed immune activation, epidermal barrier dysfunction (often linked to FLG gene mutations), and frequent associations with respiratory atopy, including asthma and allergic rhinitis. In contrast, the intrinsic phenotype, observed in the remaining 20% of patients, presents with normal IgE levels, an absence of atopic comorbidities, and a greater contribution of Th1 and Th17 immune pathways, particularly in certain ethnic populations [3,4,5].

Beyond this binary classification, emerging research highlights additional phenotypic subgroups based on age of onset, ethnicity, lesional morphology, and specific biomarker profiles. For instance, Asian AD phenotypes exhibit heightened Th17/Th22 activation with prominent lichenification, while European subtypes typically demonstrate a stronger Th2-driven response with greater xerosis and eczematous lesions. Pediatric-onset AD is often associated with systemic allergic sensitization, whereas adult-onset AD frequently involves more localized, lichenified plaques with a higher prevalence of chronic hand eczema [6,7].

These distinct phenotypic patterns underscore the complexity of AD pathophysiology and the need for precision medicine approaches to optimize treatment strategies.

Among its potential comorbidities, AD often coexists with alopecia areata (AA)—an autoimmune condition characterized by non-scarring hair loss—which may affect treatment responses and disease progression. The immunologic and inflammatory pathways shared between AD and AA suggest a complex interplay, where therapies targeting AD could influence AA outcomes in unpredictable ways. Understanding this relationship is clinically significant, especially as newer therapies for AD are explored, offering an opportunity to assess their impact on AA. This study seeks to clarify these effects, providing insights into optimal management for patients affected by both conditions [8].

Current treatments for atopic dermatitis (AD), such as topical corticosteroids, calcineurin inhibitors, and biologics targeting the JAK-STAT pathway, are increasingly being examined for their effects on autoimmune comorbidities like alopecia areata (AA). Systemic therapeutic options for AD, including conventional therapies such as systemic corticosteroids and cyclosporine, as well as novel therapies like JAK inhibitors and dupilumab, are also being explored in this context. Given the immunologic overlap in AD and AA—both characterized by Th2 and Th1/Th17 pathway activation—therapies targeting these pathways can impact AA, albeit unpredictably. JAK inhibitors, initially promising for AA, have shown varied responses when used in AD patients who also have AA, with some studies reporting improvement in AA symptoms, while others indicate minimal to no effect. Emerging evidence suggests that personalized approaches to AD treatment may help predict and optimize outcomes in patients with concurrent AA, highlighting the need for ongoing investigation into how AD therapies can be tailored for this subgroup [9,10].

Understanding the interaction between atopic dermatitis (AD) treatments and alopecia areata (AA) outcomes is clinically significant due to the shared inflammatory and immune pathways underlying both conditions. AD treatments, particularly those targeting immune modulation such as JAK inhibitors and biologics, have the potential to alter disease activity in AA—either alleviating or exacerbating symptoms depending on individual immunologic responses. This interplay is critical in clinical decision making, as optimizing treatment for AD in patients with AA may improve overall patient outcomes and reduce the risk of adverse effects from treatment overlap. Such insights are essential for developing personalized treatment strategies, addressing both the dermatologic and psychosocial impacts of these comorbid conditions.

This study aims to identify predictive factors in the progression of alopecia areata (AA) among patients undergoing treatment for atopic dermatitis (AD). By conducting a comprehensive evaluation of clinical and paraclinical features—including disease duration, individual and familial medical history, and associated comorbidities—this research seeks to uncover key variables that may forecast AA outcomes in the context of AD treatment. Through this approach, we aim to establish a basis for more informed, individualized treatment decisions that could potentially optimize therapeutic efficacy for patients with coexisting AD and AA.

## 2. Materials and Methods

This retrospective cohort study was conducted at our clinic at Railways Universitary Hospital Iasi, encompassing a population of 154 patients diagnosed with atopic dermatitis (AD). Among these, we identified 6 patients who also presented with varying degrees of alopecia areata (AA). Our inclusion criteria mandated a confirmed diagnosis of both AD and AA, allowing for a comprehensive examination across all age groups and both genders, as well as encompassing patients receiving systemic treatments.

Exclusion criteria were established to ensure the robustness of our findings. Patients who had only a single visit without subsequent follow-ups were excluded, as were those with noncompliance to prescribed treatment regimens, which could confound treatment outcome assessments. Additional exclusion criteria included patients with incomplete clinical data or those with other significant dermatological or systemic conditions that could impact the progression of AA.

Following the application of these inclusion and exclusion criteria, our final cohort consisted of 6 patients, all of whom were receiving systemic therapy. We undertook a detailed analysis of various demographic and clinical variables, including age, sex, duration of both diseases, occupational history, comorbidities, family history of autoimmune conditions, and the specific treatments administered. This multifaceted approach aims to elucidate predictive factors that may influence the outcomes of AA in the context of concurrent AD treatment.

The treatments administered to patients in this study comprised a range of biological therapies and systemic corticosteroids. Specifically, two types of Janus kinase (JAK) inhibitors were utilized: upadacitinib, which selectively inhibits JAK1, and baricitinib, which inhibits both JAK1 and JAK2. Three patients received JAK inhibitors for their AD management. Additionally, dupilumab, an inhibitor of interleukin-4 (IL-4) and interleukin-13 (IL-13), was initially administered to one patient, who subsequently switched to a JAK inhibitor after discontinuing dupilumab. Another patient received dupilumab as their sole treatment.

Furthermore, one patient was treated with systemic corticosteroids. The choice of therapies reflects the varied severity of AD among patients and aligns with contemporary treatment guidelines advocating for the use of biologics in moderate to severe cases. JAK inhibitors and IL-4/13 inhibitors have demonstrated significant efficacy in controlling inflammation and improving quality of life for patients suffering from AD, making them crucial components of our treatment regimen [11] (Table 1).

Several studies have highlighted factors that may influence the efficacy of JAK inhibitor therapy in alopecia areata, particularly the duration of severe scalp hair loss and the presence of residual scalp hair at treatment initiation. Specifically, treatment initiated within the first 3–4 years of severe hair loss has been associated with significantly improved outcomes, and patients with some residual scalp hair tend to respond better than those with near-complete or complete hair loss [8].

Although these criteria were not part of our initial patient selection process, it is noteworthy that only one patient in our cohort had a prolonged disease duration of approximately 17 years, and two patients presented with complete scalp hair loss at the initiation of therapy.

A review of the relevant literature indicates that two primary measures are commonly utilized to assess treatment outcomes in alopecia areata: the Dermatology Life Quality Index (DLQI) and the Severity of Alopecia Tool (SALT) score. The DLQI is a patient-reported outcome measure that evaluates the impact of skin conditions on quality of life, with scores ranging from 0 (no impact) to 30 (maximum impact). Lower scores indicate better quality of life. The SALT score is a clinician-reported measure that quantifies the extent of scalp hair loss on a scale from 0 (no hair loss) to 100 (total scalp hair loss).

In alignment with the prevailing consensus in the literature, we defined a positive outcome as achieving a SALT score of 20 or less and a DLQI score of 5 or less. A neutral outcome was characterized by a change of less than 10% (upward or downward) in the SALT and DLQI scores from their baseline values, and a negative outcome was defined as a worsening of these scores by more than 10%. This approach respects and builds upon existing research while allowing for standardized evaluation across our cohort [12].

## 3. Results

We evaluated the clinical and therapeutic outcomes of each patient at the 6-month mark following the initiation of their respective treatments. The mean age of the patients at the beginning of therapy was 12 years (range: 3–22 years), with a median age of disease onset of 8 years (range: 1–20 years). Among the six patients in this cohort, three were diagnosed with moderate atopic dermatitis (SCORAD scores between 25 and 50), while the remaining three exhibited severe atopic dermatitis (SCORAD > 50). This underscores that all patients included in the study suffered from at least moderate disease refractory to conventional therapies, warranting the use of advanced therapeutic interventions.

The baseline severity of alopecia areata was equally substantial. Five patients presented with severe forms of the disease, as reflected by a mean SALT score of 75.8 (range: 0–100), with three of these cases being alopecia universalis. Notably, one patient in the cohort did not exhibit alopecia areata at baseline. The impact of these conditions on patients’ quality of life was considerable, as evidenced by baseline DLQI or cDLQI scores ranging from 12 to 28, with a mean of 22.3, highlighting the significant psychosocial burden associated with these diseases (Table 2).

At the 6-month evaluation, the cohort demonstrated marked improvements. The mean DLQI score decreased from 22.3 to 2, representing a 91.03% improvement, with all patients reporting a substantial reduction in disease burden. Similarly, the mean SCORAD score decreased from 44.1 to 4.6, indicating an 89.57% improvement and reflecting a near-resolution of atopic dermatitis symptoms for most patients. The mean SALT score decreased from 75.8 to 3.3, signifying a 95.65% improvement in hair regrowth.

However, this improvement was not universal: one patient developed mild alopecia areata (SALT = 10) at 13 months during therapy with dupilumab. This condition subsequently improved under topical corticosteroid therapy. Importantly, this patient’s outcome was not captured in the 6-month analysis but is relevant for the broader evaluation of treatment dynamics.

## 4. Discussion

Treatment regimens included baricitinib (three patients), upadacitinib (one patient), dupilumab (one patient), and systemic corticosteroids (one patient). Among the six patients, five (83.3%) exhibited favorable outcomes, with significant improvements observed in both atopic dermatitis and alopecia areata. The sixth patient, while benefiting from dupilumab in terms of atopic dermatitis control, developed new-onset alopecia areata during therapy. Furthermore, another patient in the cohort, previously treated with dupilumab and later initiated on baricitinib (corresponding to patient 4 in Table 3), experienced a significant worsening of alopecia areata, ultimately leading to treatment discontinuation. Notably, this patient is the only one receiving novel therapies that did not reach a SALT score of 0 at the 13-month mark, but we attribute this to the long disease duration for this particular patient. This brings us to a total of 2 patients out of the 6 who, at some point, developed or had a worsening of alopecia areata while receiving dupilumab treatment.

These findings underscore the efficacy of advanced therapies in managing both atopic dermatitis and alopecia areata in the majority of cases. Nonetheless, the paradoxical development or exacerbation of alopecia areata in two patients receiving dupilumab raises important considerations regarding the immune-modulating effects of targeted biologics. These observations align with emerging evidence suggesting that such therapies, while effective, may unmask or exacerbate underlying autoimmune tendencies in susceptible individuals. This complexity highlights the importance of individualized treatment strategies, informed by a thorough understanding of patient-specific factors, to maximize therapeutic benefit while minimizing potential adverse effects.

According to our national treatment guidelines for atopic dermatitis (AD), particularly during flare-ups, both cyclosporine and systemic corticosteroids are considered first-line therapeutic options. However, based on our clinical experience, systemic corticosteroids are preferred due to their more rapid onset of action and superior ability to control the treatment duration. This is particularly relevant given that these therapies are recommended for short-term use in adherence to national guidelines. Moreover, corticosteroids offer enhanced flexibility in managing the dosage regimen, allowing for the administration of the lowest effective dose while minimizing the risk of adverse effects.

Janus kinase (JAK) inhibitors have emerged as a transformative therapy for alopecia areata (AA). Clinical trials have demonstrated the efficacy of baricitinib, a JAK1 and JAK2 inhibitor, in promoting significant hair regrowth in patients with severe AA. For example, two phase 3 trials demonstrated that baricitinib significantly reduced Severity of Alopecia Tool (SALT) scores at 24 weeks, showing superior efficacy compared to placebo. The safety profile of baricitinib appears favorable, with adverse events being manageable and consistent with its known effects. However, clinicians must carefully assess the risk-benefit ratio when initiating JAK inhibitor therapy, particularly in younger patients or those with comorbidities, as these agents are relatively new on the market. Long-term considerations are crucial, as discontinuation of baricitinib has been shown to frequently result in relapse of AA symptoms. A study on baricitinib withdrawal and retreatment reported that more than 80% of patients experienced symptom recurrence within 152 weeks of discontinuation, underscoring the necessity for ongoing therapy to sustain hair regrowth. This poses a challenge, as many patients, including those in our clinic, express reluctance toward life-long treatment with newer medications. Future studies should aim to establish optimal treatment durations and explore strategies to maintain remission after therapy discontinuation [13,14,15].

Recent research has further highlighted the potential role of upadacitinib, a selective JAK-1 inhibitor, in the management of patients with concomitant atopic dermatitis (AD) and alopecia areata (AA). A multicenter retrospective study assessing the effects of upadacitinib in this patient population demonstrated significant reductions in SALT scores as early as four weeks into treatment, with continued improvements over time. Notably, patients with elevated serum IgE levels and atopic comorbidities appeared to respond more favorably, suggesting that a type 2 inflammation-driven pathogenesis may influence treatment outcomes. However, nearly half of the study’s cohort did not achieve a SALT50 response, underscoring variability in therapeutic efficacy. While this study supports the use of JAK inhibitors in managing both AD and AA, it also reinforces the need for patient-specific considerations when selecting targeted therapies, as well as the importance of long-term follow-up to assess sustained disease control [16].

A recent Italian Delphi consensus provided expert recommendations on the use of Janus kinase (JAK) inhibitors in the treatment of moderate-to-severe atopic dermatitis (AD) in real-world clinical settings. The consensus highlighted the efficacy and rapid onset of action of JAK inhibitors, particularly upadacitinib and abrocitinib, in addressing both AD and associated symptoms such as pruritus. Notably, the panel agreed that JAK inhibitors may be preferred in patients with difficult-to-treat areas such as the face, neck, and hands, as well as in those with an “itch-dominant” phenotype. Despite their effectiveness, concerns about long-term safety remain, prompting recommendations for careful patient selection, especially in those with cardiovascular risk factors. The expert panel also emphasized the need for individualized patient follow-up to optimize clinical outcomes and minimize potential adverse events [17].

Real-world data further support the long-term safety and efficacy of JAK inhibitors in treating severe alopecia areata (AA). A 52-week multicenter retrospective study evaluating baricitinib in 96 adult patients with severe AA demonstrated significant clinical improvements, with 61.5% of patients achieving a SALT score ≤ 20 by the end of the study. Importantly, the safety profile of baricitinib was consistent with findings from clinical trials, with no severe adverse events (SAEs) reported. The most common mild-to-moderate adverse events included upper respiratory tract infections, headache, acne, and asthenia, while laboratory abnormalities such as hypercholesterolemia and blood cell count changes were observed but manageable. These findings reinforce baricitinib’s role as a viable and well-tolerated treatment option in real-world settings [18].

The use of dupilumab, an interleukin-4 receptor antagonist approved for atopic dermatitis (AD), in patients with concurrent AA and AD continues to raise clinical questions. While some case reports suggest that dupilumab may have a positive impact on AA symptoms, other evidence indicates the potential for dupilumab to induce or exacerbate AA. In our cohort, one patient developed mild AA after 13 months of dupilumab therapy, and another patient experienced significant worsening of preexisting AA, ultimately leading to treatment discontinuation. These findings align with emerging reports and highlight the necessity for careful patient selection and monitoring when prescribing dupilumab for patients with coexisting AD and AA [19].

An additional aspect worthy of discussion is the instance of the patient who benefited from systemic corticosteroid therapy, achieving significant improvements in both AA and AD symptoms. While this outcome was a favorable short-term result for the clinician, it is essential to emphasize that systemic corticosteroid therapy is not a sustainable long-term solution due to its significant side effect profile. Prolonged corticosteroid use is associated with numerous potential complications, including osteoporosis, adrenal suppression, hyperglycemia, hypertension, and an increased risk of infections. The cumulative burden of these adverse effects necessitates a cautious approach, reserving systemic corticosteroids for short-term interventions or selected cases where alternative therapies are not viable [20].

Nevertheless, this case serves as a reminder that conventional therapies, while not without risks, still hold a valuable place in the clinician’s armamentarium. The effectiveness of corticosteroid therapy in certain patients demonstrates that well-established treatments, grounded in decades of clinical experience, remain relevant despite the rise of newer, more targeted therapies. While innovative approaches such as biologics and JAK inhibitors offer advanced options for disease control, adherence to foundational therapeutic principles ensures a comprehensive and flexible strategy for managing complex immune-mediated conditions like AD and AA.

The coexistence of atopic dermatitis (AD) and alopecia areata (AA) presents a unique therapeutic challenge, as the interplay between these two immune-mediated conditions can influence treatment outcomes. Although current research has yet to identify specific patient characteristics that reliably predict therapeutic success, emerging evidence provides preliminary insights into potential treatment approaches.

Janus kinase (JAK) inhibitors, such as baricitinib and upadacitinib, have shown remarkable promise in addressing both AD and AA. These therapies target shared immune pathways and have demonstrated significant clinical efficacy in managing both conditions. However, questions remain regarding the sustainability of their effects, as relapse upon treatment discontinuation has been frequently observed. This raises important considerations for clinicians, particularly in cases where long-term therapy may not align with patient preferences or where the safety profile of prolonged use remains uncertain [21,22,23].

The safety and efficacy of upadacitinib have been extensively evaluated and confirmed through multiple studies, all of which reported consistent findings. Clinical trials and real-world evidence have demonstrated that upadacitinib effectively reduces disease severity in patients with atopic dermatitis (AD) and alopecia areata (AA), showing significant improvements in SALT scores and overall symptom control. Additionally, its safety profile remains favorable, with adverse events generally being mild to moderate and manageable. These consistent results across various studies reinforce upadacitinib’s role as a reliable therapeutic option for patients with AD and AA, further supporting its integration into clinical practice [24].

The role of dupilumab, an interleukin-4 receptor antagonist, is less clearly defined in patients with concurrent AD and AA. While reports in the literature highlight cases of improvement in AA under dupilumab therapy, others document paradoxical effects, including new-onset or exacerbated AA. Our clinical experience, in which two patients on dupilumab either developed or worsened alopecia areata, mirrors these conflicting outcomes and suggests the need for cautious patient selection and close monitoring when prescribing this therapy.

It is also important to acknowledge the continued relevance of conventional therapies, as demonstrated by the successful use of systemic corticosteroids in one patient from our cohort. While this approach led to notable improvements in both conditions, the well-documented side effects of systemic corticosteroids—such as adrenal suppression, osteoporosis, hyperglycemia, and heightened infection risk—limit their suitability for long-term use. Nevertheless, this case highlights that even with the advent of novel biologics and targeted therapies, established treatment options retain a critical role in specific clinical scenarios, provided their risks are carefully managed [25,26].

Overall, these findings underscore the complexity of selecting therapies for patients with comorbid AD and AA. Clinicians must navigate a delicate balance between therapeutic efficacy, patient preferences, and the potential risks associated with both conventional and innovative treatments. A personalized approach, informed by a thorough understanding of the patient’s medical history and close monitoring of treatment response, is essential. Further research is warranted to identify predictive markers and refine therapeutic strategies for this challenging patient population.

This study has several limitations that must be acknowledged. First, the small sample size, comprising only six patients, limits the generalizability of our findings to broader populations. The restricted cohort size may reduce statistical power and increase the potential for variability in observed outcomes. Additionally, the retrospective design of the study presents inherent limitations, including reliance on existing clinical records, which may not capture all relevant data with uniform precision or completeness.

Selection bias is another potential limitation, as the inclusion criteria focused on patients who presented with both atopic dermatitis (AD) and alopecia areata (AA) and who received advanced or systemic therapies. This approach may exclude patients with milder forms of the diseases or those managed with topical therapies alone, potentially skewing the findings toward more severe cases. Furthermore, the observational nature of the study precludes establishing causal relationships between treatments and outcomes, as confounding variables such as disease duration, adherence to therapy, and comorbid conditions may influence the results [27,28].

Finally, while the study offers valuable insights into the treatment of patients with concurrent AD and AA, the findings are context specific, reflecting the therapeutic practices and patient population of a single clinic. Future research with larger, prospective cohorts and standardized outcome measures is needed to validate these findings and provide more robust guidance for clinicians managing this complex patient population.

To build on the findings of this study, future research should focus on addressing its limitations and exploring unanswered questions related to the management of concurrent atopic dermatitis (AD) and alopecia areata (AA). Prospective studies with larger, multicenter cohorts are essential to establish causal relationships between therapeutic interventions and patient outcomes. Such studies could also help identify predictive factors—such as clinical, genetic, or immunologic markers—that guide the selection of optimal therapies for patients with these comorbid conditions.

Long-term investigations are needed to evaluate the sustainability of therapeutic responses, particularly for emerging treatments like Janus kinase (JAK) inhibitors. These studies should explore the safety and efficacy of prolonged therapy, as well as strategies for maintaining disease remission after discontinuation. Additionally, randomized controlled trials comparing newer therapies, such as dupilumab and JAK inhibitors, with conventional treatments like systemic corticosteroids would provide valuable insights into their relative effectiveness and safety profiles.

Further research should also address the potential paradoxical effects of therapies, particularly dupilumab, in patients with concurrent AD and AA. Elucidating the mechanisms behind these unexpected outcomes may help refine patient selection criteria and minimize adverse effects. Moreover, qualitative studies investigating patient perspectives on therapy, particularly regarding the acceptability of long-term treatment, would complement clinical research by providing insights into adherence and quality-of-life outcomes.

Finally, the integration of advanced diagnostic tools, such as dermoscopic imaging, into prospective studies could enhance our understanding of disease progression and treatment responses in this challenging patient population. At the time this article was written, the authors had evaluated possible correlations between dermoscopic findings and clinical outcomes but were unable to identify any predictive or prognostic implications. All patients in this study were evaluated dermoscopically at the beginning of the treatment and throughout its course. While the findings were consistent with universally accepted diagnostic criteria, they did not provide actionable insights for classifying patients or predicting treatment outcomes. These approaches, nonetheless, hold potential for further exploration and could contribute to a more personalized, evidence-based strategy for managing patients with both AD and AA.

## 5. Conclusions

This study highlights the intricate relationship between atopic dermatitis (AD) and alopecia areata (AA), underscoring the complexity of managing these coexisting immune-mediated conditions. Our findings demonstrate that advanced therapies, including Janus kinase (JAK) inhibitors, offer significant therapeutic potential for addressing both diseases. At the 6-month evaluation, patients treated with JAK inhibitors exhibited substantial improvements, with mean SALT scores reduced by 95.65%, mean DLQI scores improving by 91.03%, and mean SCORAD scores decreasing by 89.57%, reflecting remarkable disease control and enhanced quality of life in the majority of cases.

The study also sheds light on the paradoxical effects observed with dupilumab. While dupilumab is effective for managing AD, it was associated with either the onset or worsening of AA in two patients, suggesting the need for cautious use in individuals with these comorbid conditions. Conversely, systemic corticosteroid therapy, though limited by long-term safety concerns, demonstrated efficacy in managing both conditions in a selected patient, highlighting the continued relevance of conventional therapies in specific clinical scenarios.

Dermoscopic evaluations conducted throughout the study revealed findings consistent with established diagnostic criteria but did not offer prognostic value or aid in classifying patients or predicting treatment outcomes. These observations underscore the current limitations in using dermoscopy as a predictive tool in this context. Furthermore, based on our observations and existing literature, clinical and dermoscopic characteristics do not appear to be reliable predictors of treatment response or disease course. While dermoscopic findings aligned with established diagnostic criteria, they lacked prognostic value. Notably, elevated serum IgE levels have been suggested as a potential marker in this context, warranting further investigation [29]. 

While the authors acknowledge that immunological profiling could significantly enhance the precision of therapeutic selection, it is important to recognize that such an approach is both resource intensive and time consuming. This consideration prompted our investigation into more accessible predictive factors for treatment outcomes. In our view, two primary aspects warrant attention when selecting an appropriate therapy: first, the presence of non-atopic, non-autoimmune comorbidities, which should not be overlooked—particularly in older patients, where cardiovascular and pulmonary assessments are crucial in evaluating risk factors for complications associated with systemic corticosteroid therapy and novel therapies (especially JAK inhibitors); and second, atopic and autoimmune (or immune-mediated) diseases, as we attempted to illustrate in this work. Clinically, these findings emphasize the necessity of a personalized, patient-centric approach to treatment selection, informed by the interplay of disease severity, therapeutic efficacy, and the potential risks of adverse effects. While novel therapies provide transformative options for disease management, the integration of conventional approaches remains vital in achieving optimal outcomes in this complex patient population.

## Figures and Tables

**Table 1 diagnostics-15-00520-t001:** Baseline characteristics of patients included in our cohort. (* BMI not applicable to 2 patients).

Items	*n* = 6
Sex, *n* pts (%)	
Male	2 (33.3)
Female	4 (66.6)
Age [median (IQR)]	12 (2–22)
BMI * [median (IQR)]	22.3 (19.8–24.5)
Total serum IgE level (patients)	
>100 IU/mL [*n* (%)]	2 (33.3)
≤100 IU/mL [*n* (%)]	4 (33.3)
AD family history, *n* pts (%)	
Yes	3 (50)
No	3 (50)
AA localization, *n* pts (%)	
Scalp (fronto-parietal)	6 (100)
Scalp (ophiasis)	4 (66.6)
Eyebrows	3 (50)
Trunk and limbs	2 (33.3)
Universalis	4 (66.6)
Mean AA duration, months ± SD	46 ± 61.34
Median age of AA onset [median (IQR)]	8 (1–20)
Concomitant autoimmune disease, *n* pts (%)	
No	3 (50)
Graves’ disease	1 (16.6)
Hashimoto thyroiditis	2 (33.3)
Atopic Comorbidity, *n* pts (%)	
Yes	2 (33.3)
No	2 (33.3)
Asthma	1 (16.6)
Rinithis	1 (16.6)
Food allergy	1 (16.6)
Conjunctivitis	1 (16.6)
Previous systemic therapies for AD, *n* pts (%)	
No	1 (16.6)
Oral corticosteroids	5 (83.3)
Dupilumab	1 (16.6)
Severity scores assessed at baseline [median (IQR)]	
SALT score	77.5 (65–100)
SCORAD	44.1 (31–63.6)
DLQI score	22.3 (12–28)

**Table 2 diagnostics-15-00520-t002:** The evolution of relevant severity scores for each patient included in the study, 6 months into the treatment.

	SALT Score		DLQI/cDLQI		SCORAD	
	Baseline	6 Months	Baseline	6 Months	Baseline	6 Months
Patient 1	100	0	28	2	50.3	3
Patient 2	65	0	25	1	47	0
Patient 3	100	0	25	1	47	0
Patient 4	100	10	21	1	36	0
Patient 5	0	0	26	0	63.6	5
Patient 6	90	20	12	0	38	12
Mean	75.8	5	22.3	2	44.1	4.6
Improvement	92.08%		91.03%		89.57%	

**Table 3 diagnostics-15-00520-t003:** The evolution of SALT score at 13 months into the treatment. N/A refers to patients who have yet to undergo this length of therapy. Another relevant information to be taken out of this table is the maintenance of patient 2’s perfect score value.

	SALT Score			Age	Drug Administered
	Baseline	6 Months	13 Months		
Patient 1	100	0	N/A	4	baricitinib
Patient 2	65	0	0	22	baricitinib
Patient 3	100	0	N/A	17	upadacitinib
Patient 4	100	10	N/A	19	baricitinib
**Patient 5**	**0**	**0**	**10**	**3**	**dupilumab**
Patient 6	90	20	N/A	8	systemic corticosteroids
Mean	75.8	5	N/A		
Improvement		92.08%	N/A		

## Data Availability

The original contributions presented in this study are included in the article. Further inquiries can be directed to the corresponding authors.

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
