# Peer review of "Dual Relief: How Atopic Dermatitis Treatments Affect Alopecia Areata—A Small Retrospective Cohort Study"

_diagnostics, 2025, doi:10.3390/diagnostics15050520_

Round 1

Reviewer 1 Report

Comments and Suggestions for Authors

It is an interesting study on a topic that doctors who care for patients with AD and AA are confronted with. In the age of precision medicine, it is important to draw attention to this issue, but one study with 6 patients is not enough to adequately address this issue, which is also cited as a limitation by the authors. In my opinion, the decision  which drug should be chosen as the optimal therapy for patients with AD and AA cannot be made on the basis of dermatoscopic or other clinical examinations. Rather, the immunological profile and endotypes should be defined in order to ultimately use the right medication. Although the authors did not work on this, this point should be addressed in the discussion.

Author Response

We extend our gratitude for your time and attention. We are glad to receive your observations and we'd like to communicate our point-by-point responses:

Comment 1: In the age of precision medicine, it is important to draw attention to this issue, but one study with 6 patients is not enough to adequately address this issue, which is also cited as a limitation by the authors.
Response 1: Yes, unfortunately for our research but fortunately for our patients, not many times has AA occurred as a further burden. We'd like to expand our observations by analysing more such cases, hopefully in the future we will gain access to more cases that allow us to strengthen or modify our views. We also consider this our primary limitation for the study. 

Comment 2: In my opinion, the decision  which drug should be chosen as the optimal therapy for patients with AD and AA cannot be made on the basis of dermoscopic or other clinical examinations. Rather, the immunological profile and endotypes should be defined in order to ultimately use the right medication. Although the authors did not work on this, this point should be addressed in the discussion.
Response 2: Thank you for pointing this out. Indeed, this is a conclusion we also came to after scrutinizing this lot. We have modified the article to clarify and underline this observation and included other possible directions that would represent better options for optimising the therapy choosing process. Therefore, we modified the conclusions section to include this point, which can be seen in lines 445-469 of the second version of the manuscript.

Reviewer 2 Report

Comments and Suggestions for Authors

The paper is an interesting clinical observation confirming the effectiveness of biological therapy in AD patients who additionally suffer from AA.  Both diseases can coexist and also present similar pathogenetic mechanisms. Treating them together would be an fascinating therapeutic option.

However, it should be emphasized that cyclosporine, a conventional drug used in both AD and AA treatment, has also a similar therapeutic effect.

This is a retrospective study in which a group of 6 people in whom AA was observed, was selected from 154 patients with atopic dermatitis.

The small size of the group does not allow us to draw general conclusions, but it certainly indicates a potential research direction. Table 3 is largely a repetition of the information contained in Table 2. Therefore, I would suggest expanding it with information about the drug that a particular patient received and the age of the patient. Did a patient who developed alopecia during AD treatment receive Dupilumab? Was the second patient with AA exacerbation also treated with dupilumab? This was not clearly stated in the Results section.

Another advantage of the study is the authors' critical assessment of the duration of the therapeutic effect after the use of JAK inhibitors. We do not have an answer to this question yet.

The work is interesting, written in good language, and the literature is selected in the right way.

Author Response

We'd like to firstly extend our gratitude for your time, attention and appreciation of our work. We are delighted you find the piece interesting and well done. We'd like to respond to your observations:

Comment 1: However, it should be emphasized that cyclosporine, a conventional drug used in both AD and AA treatment, has also a similar therapeutic effect.
Response: In accordance with our national treatment guidelines for atopic dermatitis (AD), particularly during flare-ups, cyclosporine and systemic corticosteroids are both considered first-line therapies. However, based on our experience, we have found that systemic corticosteroids tend to provide a quicker response and offer better control over treatment duration. This is important, as these therapies are recommended for short-term use according to our guidelines. Additionally, corticosteroids allow for more precise management of the dosage regimen, enabling us to prescribe the lowest effective dose while minimizing the risk of side effects. However, we acknowledge the fact that cyclosporine should be mentioned in this article, which is why we modified our manuscript - lines 76-78 now mention cyclosporine and lines 219-227 now clarify the reason why cyclosporine was less discussed in this work.

Comment 2: Table 3 is largely a repetition of the information contained in Table 2. Therefore, I would suggest expanding it with information about the drug that a particular patient received and the age of the patient. 
Response 2: We have made the suggested modification. Thank you for pointing this out!

Comment 3:  Did a patient who developed alopecia during AD treatment receive Dupilumab? Was the second patient with AA exacerbation also treated with dupilumab? This was not clearly stated in the Results section.
Response 3: We have indeed poorly explained this and have made the modification to clear out the events, as seen in lines 203-214 of the revised manuscript.

Reviewer 3 Report

Comments and Suggestions for Authors

This article could be of some interest. The main limitation is the small sample size.

The discussion should be enhanced, as several reports have been published on the use of JAK inhibitors in patients with both AA and AD. In particular, I would suggest reading this multicentre study (10.1016/j.jaad.2023.05.001) and this Delphi consensus of the use of JAK inhibitors: 10.1007/s13555-024-01135-x.

Regarding JAK inhibitors, long-term data are available in clinical practice nowadays. The discussion should cite these works, since RWE are crucial. For baricitinib, data are now available in real world clinical practice after 52 weeks of follow up in AA patients (10.1080/09546634.2024.2444494). Also, for upadacitinib there are long-term data in AD in clinical practice (10.1007/s13555-024-01334-6. ; 10.1111/jdv.19507)

Upadacitinib, dupilumab baricitinib should not be spelled with the capital letter.

EASI scores should be provided if available

Author Response

Thank you for taking the time to revise our work. Your feedback is greatly appreciated and valued. 

We have read the recommended articles and modified the information in our manuscript in concordance with the materials you provided. We thank you for this contribution! You can find the modifications in rows 274-296 of our updated manuscript, as well as 347-355.

Unfortunately, we are unable to provide EASI scores as they are not routinely used in Romania.

We have rectified the mistake of having spelt upadacitinib and other therapies with a capital letter. Thank you for pointing this out!

Round 2

Reviewer 3 Report

Comments and Suggestions for Authors

Thank you for revising the manuscript, I think it has been improved

I have no further queries for this manuscript